# Interactions between Dental MSCs and Biomimetic Composite Scaffold during Bone Remodeling Followed by In Vivo Real-Time Bioimaging

**DOI:** 10.3390/ijms24031827

**Published:** 2023-01-17

**Authors:** Ana Catarina Costa, Patrícia Mafalda Alves, Fernando Jorge Monteiro, Christiane Salgado

**Affiliations:** 1Instituto de Investigação e Inovação em Saúde (i3S), Universidade do Porto, Rua Alfredo Allen, 208, 4200-135 Porto, Portugal; 2Instituto Nacional de Engenharia Biomédica (INEB), Rua Alfredo Allen, 208, 4200-135 Porto, Portugal; 3Faculdade de Engenharia, Universidade do Porto, Rua Dr. Roberto Frias, s/n, 4200-465 Porto, Portugal; 4Faculdade de Medicina Dentária, Universidade do Porto, Rua Dr. Manuel Pereira da Silva, 4200-393 Porto, Portugal; 5Porto Comprehensive Cancer Center (P.CCC), R. Dr. António Bernardino de Almeida, 4200-072 Porto, Portugal

**Keywords:** collagen, nanohydroxyapatite, osteopontin, dental follicle MSC, biomaterial

## Abstract

Oral–maxillofacial tumor removal can generate critical bone defects and major problems for patients, causing dysfunctionalities and affecting oral competencies such as mastication, swallowing, and breathing. The association of novel biomaterials and cell therapies in tissue engineering strategies could offer new strategies to promote osteomucosa healing. This study focused on the development of a bioengineered construct loaded with human dental follicle cells (MSCs). To increase the bioconstruct integration to the surrounding tissue, a novel and comprehensive approach was designed combining an injectable biomimetic hydrogel and dental stem cells (hDFMSCs) expressing luminescence/fluorescence for semi-quantitative tissue imaging in live animals. This in vivo model with human MSCs was based on an intramembranous bone regeneration process (IMO). Biologically, the biocomposite based on collagen/nanohydroxyapatite filled with cell-loaded osteopontin–fibrin hydrogel (Coll/nanoHA OPN-Fb) exhibited a high cellular proliferation rate, increased bone extracellular matrix deposition (osteopontin) and high ALP activity, indicating an early osteogenic differentiation. Thus, the presence of human OPN enhanced hDFMSC adhesion, migration, and spatial distribution within the 3D matrix. The developed 3D bioconstruct provided the necessary pro-regenerative effect to modulate the biological response, precisely fitting the bone defect with fine-tuned adjustment to the surrounding original structure and promoting oral osteomucosa tissue regeneration. We were also able to track the cells in vivo and evaluate their behavior (migration, proliferation, and differentiation), providing a glimpse into bone regeneration and helping in the optimization of patient-specific therapies.

## 1. Introduction

Oral cancers are mostly derived from the mucosal epithelium in the oral cavity, pharynx, and larynx and are known collectively as head and neck squamous cell carcinoma (HNSCC) [1] with an incidence of around 6 per 100,000 people worldwide (new cases: 377,113/year, deaths: 177,757 in 2020), estimated to rise to 47% by 2040 [2]. The screening strategy for HNSCC has not proved to be effective, and pre-malignant lesions usually progress to invasive cancer [1] with the bone invasion rate reaching 58% [3]. Clinically, patients are generally treated with surgical resection, followed by adjuvant radiation or chemoradiation [1], after which 66% remain with disfigurements and scars, requiring restoration of large bone defects [4]. The gold standard treatment after surgical tumor removal remains the use of bone autografts, particularly to treat and reconstruct large segmental bone defects [5]. Autografts have osteogenic, osteoconductive, and osteoinductive properties, but there are some disadvantages such as the risk of vascular–nervous lesions and increased patient morbidity (second surgery) [6]. Moreover, within the first year of surgery, the failure rate of the autografts (above 5 cm) can reach 75% [7]. Commercially available products, such as metallic [8] and ceramic [9] materials, usually have a lack of the complex bone tissue structure (architecture and porosity) because of their stiffness and processing requirements, resulting to an inferior bone tissue integration and repair. Another limitation of commercial biomaterials is the possible release of toxic ions and/or particles that lead to an inflammatory response, which may reduce biocompatibility and cause implant loss [8,9]. The association of novel biomaterials and cell therapies in tissue engineering strategies could offer new approaches to promote osteomucosa healing. This study focused on the development of an osteoinductive hydrogel loaded with human dental MSC and injected inside a 3D composite construct. We also aimed at using an ectopic bone remodeling model with a novel, highly sensitive optical imaging approach allowing real-time in vivo human cell tracking in the bone tissue regeneration process into an implanted engineered biomaterial. This bioengineering approach can be applied to the bone tissue and used as a personalized therapy making use of the patient’s adult dental stem cells, which are a cornerstone of regenerative therapy.

In the oral cavity, adult tooth tissues contain different active populations of stem cells with a mesenchymal phenotype (DMSCs). These stem cells can be isolated from different parts of the dental tissue, and their main advantages are their easy access with limited morbidity, multipotent capacity, immunosuppressive properties, rapid attachment, and high proliferation rates [10,11,12]. However, improved knowledge about soluble signals involved in osteogenic differentiation is still required. Furthermore, these MSCs require an efficient 3D scaffold material that can facilitate their integration, differentiation, matrix synthesis, and mineral deposition. Porous biocomposite scaffolds based on collagen/nanohydroxyapatite (Coll/nanoHA) have shown to have excellent results in bone tissue regeneration applications [13,14]. One major challenge in bioengineered constructs concerns cell delivery, namely, ensuring the functionality of administered cells within the host environment, without premature loss and/or excessive death. In this context, an osteopontin and fibrin-based hydrogel may have a key role, as it can modulate the cellular dynamics and regulate cell organization and morphogenesis. Osteopontin (OPN) represents 1–2% of the non-collagenous proteins present in the bone matrix, consisting of about 297 amino acids [15]. OPN is a highly charged and phosphorylated protein characterized by a fairly high affinity to calcium. OPN also could modulate the nucleation of calcium phosphate during bone mineralization, and regulate the growth, shape, and size of HA crystals [16,17,18]. The presence of OPN attracts osteoblasts to the bone resorption sites and promotes bone matrix deposition to fill the cavity. For this reason, OPN was used as a cytokine within a fibrin gel to mediate the attachment and communication between mesenchymal stem cells and/or osteoblasts [18,19]. Fibrin is described as a natural fibrous protein that possesses distinct advantages such as biocompatibility and injectability (capable of filling any shape or geometry of a gap) and contains numerous specific bonding sites for cell–matrix interactions, promoting cell infiltration and proliferation. However, it has low mechanical properties and a high enzymatic degradation rate, so it has been used as a cell delivery biomaterial [20,21,22]. 

The scaffold should replace the damaged/lost bone tissue, promoting new tissue ingrowth, while avoiding excessive inflammatory reactions after implantation and foreign body response [23]. Macrophages play a critical role in the biomaterial-induced inflammatory response since they secrete important signaling molecules [24] and can be polarized (M1 or pro-inflammatory and M2 or anti-inflammatory) and changed depending on the cytokine level or other soluble factors in the microenvironment (e.g., exosomes with miRs, cricRNAs, and lncRNAs) [25]. The assessment of the immune cells’ activation and migration can enable the evaluation of the interaction between the remodeling process and the immune system and should mimic the host response cascade following the 3D bioconstruct after in vivo implantation [26,27].

In mapping the development of bone in the tissue-engineered model, cells should be tracked to provide a quantitative glimpse into the efficiency of the cell therapy and provide specific optimization (cell number, viability, proliferation rate) of patient-specific therapies. To that end, luminescent/fluorescent optical imaging in living animal subjects allows the imaging and characterization of transplanted cells that could offer many advantages, such as continuous monitoring and 2D visualization of tissues in living animals, thus reducing the number of subjects required for the research [28] since luminescence signals residing deep within tissues can be detected with excellent sensitivity (without animal tissue background) [29]. With that real-time imaging purpose, GFP-Luc plasmid was transfected to hDFMSC cells via a lentivirus vector for luminescent cell detection in vivo. Cells permanently expressing luminescence/fluorescence can provide an advantage over normal cell labeling systems because the luminescent/fluorescent label is integrated into the genome and as a result is retained in the daughter cells, making it possible to follow cell proliferation, without requiring co-injection of imaging agents, providing a more accurate signal to a location inside an animal subject and enabling it to be followed over time [30,31]. 

The association of novel biomaterials and cell therapies in tissue engineering therapy could offer new strategies to promote maxillofacial bone healing. To address this objective, a 3D bioconstruct based on a collagen/nanohydroxyapatite porous scaffold (Coll/nanoHA) filled with a hydrogel of fibrin modified with osteopontin (OPN-Fb) to carry dental MSCs was developed. This combined approach (scaffold + cell-loaded hydrogel) can improve the mechanical properties of the graft and provide the necessary pro-regenerative effect to modulate the biological response, and it can precisely fit the bone defect fine-adjusted to the surrounding original bone structure.

## 2. Results

### 2.1. Mechanical and Rheological Behavior

#### 2.1.1. Viscoelastic Analysis of Osteopontin–Fibrin Hydrogel (OPN-Fb)

The OPN–fibrin hydrogel was developed to carry cells and also be injected into a 3D Coll/nanoHA porous scaffold in order to improve the mechanical strength of the biomaterial and enhance bone tissue regeneration. The rheological characteristics of the OPN-Fb hydrogel were studied by rheometer analysis. The results showed the storage modulus (G′) that referred to the elastic behavior of the gel. Thus, a higher G′ modulus reflects an increase in the hydrogel stiffness. Figure 1a shows the storage modulus (G′) for a low- (0.1 Hz) and high-frequency value (1 Hz). A two-way ANOVA test was performed for these two different frequency ranges and no statistical difference was observed, but there was a tendency for an increase in the G′ modulus for higher frequency values of the fibrin hydrogels. The OPN influence on the rheological properties of the hydrogels was analyzed at high (HC—100 µg/mL) and low concentrations (LC—10 µg/mL). At a higher frequency, the results show a remarkably higher G′ modulus for fibrin gel when compared with the hydrogels in the presence of OPN protein at high and low concentrations. 

The loss modulus (G″) of the samples with OPN LC and HC is displayed in Figure 1b. When compared to the results of the G′ modulus (Figure 1a), the G″ modulus of the samples presents a similar behavior for all the samples with or without OPN. A two-way ANOVA statistical test was performed for the two different frequency ranges and no statistical difference was found. The ratio of G″ and G′ (tan δ) values is shown in Figure 1c for the samples with OPN (low and high concentration). The Figure 1c shows that with increasing frequency the hydrogel samples in the presence of OPN protein tended to become more viscous and less elastic when compared with the fibrin sample. At the lowest frequencies, the samples showed a more elastic behavior, and there were no differences between all the samples, with or without OPN.

#### 2.1.2. Mechanical Characterization of the Coll/nanoHA Scaffold Loaded with OPN-Fibrin Gel

DMA analyses were performed to better understand the mechanical influence of OPN protein incorporated within the fibrin hydrogels and injected into the Coll/nanoHA scaffolds. Figure 2a shows the changes in storage modulus (E′) with increasing frequency for the samples with low OPN concentration (LC) and high concentration (HC). As can be seen in the graph, the inclusion of OPN protein increased the viscoelastic properties of the Coll/nanoHA composite with a high OPN concentration (100 μg/mL). For the composite with a lower OPN concentration (10 μg/mL), the rise in the E′ modulus with the frequency was lower when compared with only the fibrin hydrogel.

Figure 2b shows an increase in the E′′ modulus with the increase in frequency for all the samples. Coll/nanoHA OPN-Fb (HC) showed the highest E′′ modulus, followed by Coll/nanoHA Fb. However, both materials showed similar behavior (2.3 × 10^4^ ± 1.5 × 10^4^ Pa and 3.4 × 10^4^ ± 1.5 × 10^4^ Pa, respectively) with 10 Hz of frequency.

Therefore, the highest OPN concentration led to an increase in mechanical properties, and in contrast the lower OPN concentration caused a decrease in the E′′ (lower stiffness).

The loss factor (tan δ), representing the ratio between the E′′ and E′ modulus, is shown in the graph in Figure 2c and reflected a tendency for all composites to become more viscous and less elastic with the increase in frequency. Therefore, the OPN protein incorporation in the Coll/nanoHA Fb composite had a significant influence on the mechanical properties (higher stiffness), but there were no statistical differences between samples with different OPN concentrations.

### 2.2. In Vitro Biological Studies

#### 2.2.1. DNA Quantification Assay

The hDFMSC proliferation rate within the Coll/nanoHA Fb, Coll/nanoHA OPN-Fb, and Coll/nanoHA 3D composite scaffolds was estimated by DNA extraction quantification shown in Figure 3. It was possible to observe that hDFMSC exhibited the highest overall proliferation rates in the Coll/nanoHA OPN-Fb biocomposite when compared with all the other samples. Moreover, after 14 and 21 days of culture, total DNA content was three- and two-fold higher, respectively, when compared with 7 days of culture. A different proliferation rate was observed in Coll/nanoHA Fb with a decrease in the DNA concentration from day 7 to day 14 of culture. In the same material, from day 14 to 21 a slight increase was noticed, although it was not statistically different. The Coll/nanoHA scaffold showed similar results during all times of culture, with a non-statistical difference, but a slight increase was observed in the total DNA content from day 14 to 21.

#### 2.2.2. Cellular Differentiation Assay

The osteogenic differentiation of hDFMSC within Coll/nanoHA Fb, Coll/nanoHA with OPN-Fb, and Coll/nanoHA biocomposite scaffolds was assessed by measuring the alkaline phosphatase enzyme (ALP) activity during 21 days of cell culture. ALP is present as a membrane marker for all types of stem cells, and it is also an earlier marker of osteogenic differentiation. A two-way ANOVA statistical test was performed for the different samples and time points, as shown in Figure 4a. Statistical differences in ALP activity were observed for hDFMSC cultured within Coll/nanoHA with the OPN-Fb hydrogel and showed an increase in the enzyme activity after 7 days, regarding an early stage of differentiation, and after 21 days regarding a later osteogenic maturation. Coll/nanoHA and Coll/nanoHA Fb samples followed the same pattern, although the Coll/nanoHA Fb showed a small decrease in ALP activity at day 14. 

The total protein content of the Coll/nanoHA Fb, Coll/nanoHA OPN-Fb, and Coll/nanoHA scaffolds with hDFMSC cultured for 7, 14, and 21 days is shown in Figure 4b. Statistical differences were observed between Coll/nanoHA Fb and Coll/nanoHA from day 7 to 21 and Coll/nanoHA with OPN-Fb for all time points. Moreover, higher protein concentrations were observed within the Coll/nanoHA OPN-Fb scaffold presenting a peak at day 14, followed by a decrease at a later time point. The Coll/nanoHA Fb and Coll/nanoHA biocomposite scaffolds showed a decrease in the total protein content between early time points of culture (7 days) and the later ones (21 days).

The cells seeded inside the scaffolds were also analyzed by RT-qPCR for the expression of relevant osteoblastic genes, using glyceraldehyde 3-phosphate dehydrogenase (GAPDH) as a housekeeping gene. Figure 5 shows different cell gene expressions (OPN, BMP-2, Oct3/4, and DMP-1) within the different scaffolds (with or without fibrin and OPN gel). In terms of the OPN expression, there is a tendency for higher gene expression with the presence of OPN in the hydrogel, although these results are not statistically different. Nevertheless, there are significant differences between Coll/nanoHA OPN-Fb and the other scaffolds without OPN regarding the BMP-2 expression (** *p* < 0.01). The hDFMSC cultured within Coll/nanoHA showed higher expression of Oct3/4 (* *p* < 0.05), an MSC pluripotency marker, when compared with the cells-loaded and injected into the hydrogel (with or without OPN). 

#### 2.2.3. Confocal Laser Scanning Microscopy

The cellular morphology and the human proteins secreted by hDFMSC were evaluated by the staining of cell actin cytoplasm and human OPN within Coll/nanoHA Fb, Coll/nanoHA OPN-Fb, and Coll/nanoHA scaffolds after 7, 14, and 21 days of culture. CLSM images in Figure 6 show that in the Coll/nanoHA OPN-Fb scaffold, hDFMSC, in the early stages of cell culture, had a lower number of cells spread out and lower accumulation of OPN protein. However, at later stages (21 days), a higher number of cells adhered along the material’s surface with aggregates on the edges and also higher deposition of human OPN, creating a continuous peripheral layer. Furthermore, the CLSM images also show that on the Coll/nanoHA Fb and Coll/nanoHA OPN-Fb 3D biocomposite scaffolds, at 7 and 21 days of culture, the human OPN protein was well distributed and seemed to follow the irregularities of the materials’ surfaces in accordance with the higher number of cells. Conversely, on both scaffolds, at day 14, a decrease in OPN presence and the aggregates of cells was observed.

The hDFMSC morphology and distribution within the Coll/nanoHA Fb, Coll/nanoHA OPN-Fb, and Coll/nanoHA (without hydrogel) scaffolds after 7, 14, and 21 days of culture were observed using CLSM (Figure 7). The pore walls of all samples are covered by the cell monolayer, which seems to follow the irregularities of the materials’ surface, as illustrated in Figure 7. CLSM images also show that hDFMSCs seeded on the Coll/nanoHA OPN-Fb biocomposite scaffold were well spread, entirely covering the scaffold surfaces. In addition, at a later time point of culture, it was possible to observe the highest number of cells creating a continuous cell layer over the material’s surface. Thus, at the 21st day of culture, the Coll/nanoHA OPN-Fb scaffold showed the highest cell density (cellular aggregates). On the contrary, the Coll/nanoHA Fb and Coll/nanoHA scaffolds had a lower number of cells and presented a more wide-spread distribution. Moreover, it was clear that at early time points the cells displayed a spread and well-adhered morphology, essentially within the Coll/nanoHA Fb and Coll/nanoHA scaffolds, and at the latest time points of culture hDFMSC became rounder and agglomerated, mainly within the 3D biocomposite modified with OPN.

#### 2.2.4. Assessment of Immune Cell Activation (Macrophages Polarization—M1 and M2)

In order to determine the role of hDFMSC within different biomaterials on the immune response that could be relevant to bone tissue engineering applications, macrophages were treated with several stimuli (direct scaffold contact or hDFMSC secretome). We measured the macrophage polarization phenotypes status to M1 (pro-inflammatory) or M2 (anti-inflammatory). Those immune cells polarization was evaluated by flow cytometry for cell surface expression of the M1 marker CD86 and M2c marker CD163. The results show a significantly higher percentage of cell surface marker CD86 in secretome-treated (basic medium, 28.3%) macrophages when compared with the non-treated macrophages and those treated with osteoinduction secretome (2 and 17.2%, respectively, Table 1, Figure 8a). Similar results were found between secretome-treated (basic medium) and INFγ-treated cells (M1 polarization, Figure 8a). 

With higher macrophage polarization results, secretome-treated (basic medium) macrophages showed significantly improved levels of CD163 (49.6%) when compared with non-treated macrophages and osteoinductive secretome (0.8% and 19%, respectively, Table 1). With a lower number of cells, the macrophage polarization results with secretome-treated (basic medium) and IL10-treated cells (M2 polarization, Figure 8a) were similar (Table 1, Figure 8b). 

Therefore, similar results were not observed with the cells cultured within the scaffolds in direct contact with the macrophages (basic and osteoinductive medium) for M1 polarization (16.2 and 17.7%, respectively, Table 1 and Figure 8c) and M2 polarization (16.3 and 14.5%, respectively, Table 1, Figure 8d). There was a slight M1 and M2 polarization when compared with the indirect contact (secretome).

### 2.3. In Vivo Biological Evaluation

The mice model of subcutaneous implants for ectopic bone formation should allow for the evaluation of transplanted human cells in terms of viability, proliferation, migration, and osteogenic differentiation capacity. For this study, a 4-week duration period was the time selected to assess the cell survival, proliferation, and extracellular matrix deposition capacity (cellular differentiation). The histological findings support that the cell-loaded scaffold enhanced animal tissue ingrowth and angiogenesis. Importantly, luminescent cells were detected for 4 weeks (Figure 9), and after being explanted the implants showed newly formed connective tissue and neo-angiogenesis within the scaffold (Figure 10A–D).

After immunostaining the human cells inside the implants, it could be observed that in all the cell-loaded scaffolds implanted for 4 weeks, small amounts of GFP-Luc hDFMSCs were still prevalent on the materials’ surfaces (Figure 10G,H). The human proteins secreted by the dental MSCs were evaluated by the detection of human OPN inside the scaffold (Figure 10E,F). Both GFP+ MSCs showed OPN accumulation at the periphery of the scaffold (Figure 10F), in accordance with the more significant presence of GFP-Luc hDFMSC in that region (Figure 10G,H). Comparing the results, the presence of OPN was more evident in hDFMSC-loaded scaffolds (Figure 10E).

## 3. Discussion

Major oral bone defects are associated with trauma, osteonecrosis, tumor removal, and congenital disorders. Over the past decades, bone tissue engineering has arisen as a powerful tool for developing new bone treatment options to overcome shortages of existing bone graft materials and bone disease therapies. To address this challenge, the main goal of this study was to develop a biomimetic 3D Coll/nanoHA porous scaffold containing tooth-derived stem cells (from dental follicle tissue) loaded into an OPN-Fb hydrogel in order to fully regenerate bone tissue and improve the early recovery of patients with critical maxillofacial bone defects. 

Biologically, the increase in the stiffness of the hydrogel has a direct impact on cell behavior. Previous studies showed that a high concentration of fibrinogen and thrombin resulted in a denser structure, affecting the normal cell behavior [22]. Thus, lower concentrations of fibrinogen promoted an increase in cell proliferation, while a high concentration of fibrinogen promoted an increase in cell differentiation, but, adversely, could decrease the final mechanical properties (e.g., compressive strength) [32]. A rheological assay was performed to evaluate the influence of OPN protein on the viscoelastic properties of the fibrin hydrogels. The rheometer analysis offers a rheological characterization of the material, studying the deformation and flow of the samples under an applied force. Therefore, an oscillatory shear stress force is applied to the samples displaying the mechanical characteristics of the materials (i.e., shear storage modulus (G′) and shear loss modulus (G′′) [33]. The OPN-Fb hydrogel at 10 μg/mL (LC) and 100 μg/mL (HC) showed similar rheological characteristics, evidencing that OPN protein did not negatively affect the mechanical response of the composite. According to the literature review, the OPN protein is crucial in the modulation of osteoblasts, osteoclastic function, and matrix mineralization [34,35]. Previous studies have shown that the OPN protein is crucial to promote the attachment of HA crystals, favoring the osteoblast’s and osteoclast’s adhesion, and subsequently promoting their growth and proliferation. Once this protein affects cellular function, the inclusion in fibrin hydrogel is a key aspect in mimicking ECM outside of the confined tissue matrix. The normal OPN concentration in plasma blood is similar to the lower concentration (10 μg/mL) used in this study [33,35,36]. 

The biomechanical/rheological properties of the composite (Coll/nanoHA scaffold + fibrin hydrogel with and without OPN) were tested by dynamic mechanical analysis (DMA) assay. DMA assay is a simple technique that is used to measure the physical properties (e.g., storage modulus and loss modulus) by applying an oscillatory force to a sample and analyzing the response of the material. The storage modulus (E′) or elastic modulus describes the elastic mechanism of the material in storing energy to a predefined strain. The DMA results allow the measurement of the stiffness of the samples. Thus, the higher the E′ modulus, the higher the stiffness of the material [37]. This technique is crucial for bone tissue engineering, allowing the simulation of the in vivo cyclic loading. Thus, this assay facilitates the study of the mechanical integrity of the scaffolds, evaluating their biomechanical capability to provide temporary mechanical support to the host site. The DMA results showed that the incorporation of fibrin hydrogels with Coll/nanoHA scaffolds resulted in an improvement in the final mechanical strength of the scaffold. The E′ and E″ moduli of the hydrogel samples with a lower OPN concentration were very similar to those of the hydrogel without OPN, indicating that this protein did not influence the rheological properties of the composite. A previous study had shown that the inclusion of fibrin hydrogels within porous materials resulted in an improvement in the final strength of the composite (scaffold + fibrin gel) [38]. In a study by Brougham et al. [39], fibrin hydrogel was incorporated into the matrix of a collagen and glycosaminoglycan (CG) scaffold, prepared using a lyophilization process. In that study, it was possible to observe that the mechanical properties of the CG scaffold increased with the incorporation of fibrin hydrogel. The compressive modulus of fibrin hydrogel, of only 0.49 ± 0.1 kPa, increased to 2.97 ± 0.5 kPa with the reinforcement of the CG scaffold. Another study by Foroushani et al. [40] showed that incorporating a gelatin–glycosaminoglycan (GG) electrospun matrix into a fibrin hydrogel provided an increase in the final mechanical properties. The Young’s modulus and tensile strength of the fibrin gel with the GG scaffold showed a significant increase of 0.662 ± 0.046 MPa and 0.886 ± 0.061 MPa, respectively. The fibrin hydrogel injected into the Coll/nanoHA scaffold showed higher mechanical compressive strength. As a result, the scaffolds became stiffer while keeping their elastic behavior. This composite showed the highest E′ and E′′ moduli when compared with the other tested composite materials. The tan δ results (Figure 2c) indicated that Coll/nanoHA scaffolds with fibrin hydrogel became more viscous and less elastic with increasing frequency. Thus, the composite under compressive stress had a viscoelastic behavior with higher stiffness at higher frequency when compared with lower ones, thus leading to an appropriate behavior for the regeneration of maxillofacial bone tissue. 

The application of hDFMSC in 3D biomimetic scaffolds with the purpose of bone tissue engineering has not been totally explored yet. Since bone-marrow-derived MSCs present a significant age-related decrease in terms of differentiation potential and frequency [41], the use of human dental mesenchymal cells, in particular hDFMSC, has raised interest in regenerative medicine because they can be isolated from extracted impacted third molars, which are usually discarded as dental medical waste, without requiring an extra surgery [42]. Furthermore, these neural crest-originated cells display other superior characteristics, namely, easy accessibility, a high viability and proliferation rate, active self-renewal capability, immunomodulatory properties, feasible cryopreservation, and the absence of ethical-related issues [41,43,44,45].

To clarify the cellular basis of tissue regeneration, evaluating the multipotential capabilities of stem cells to differentiate into the desired target tissue is vitally important [46]. hDFMSCs harbor a multipotent differentiation capacity with high pluripotency and plasticity, as they can differentiate into osteoblasts, chondrocytes, adipocytes, cardiomyocytes, hepatocytes, neuronal cells, fibroblasts, cementoblasts, salivary gland cells, ductal cells, and periodontal ligament cells [43,45]. According to Rezai-Rad et al. [42], DFSCs have a strong osteogenic capability to differentiate toward the osteoblastic lineage. Graziano et al. [46] confirmed that dental mesenchymal cells are a promising source for bone tissue regeneration due to their high capacity to adhere to biomaterial surfaces.

Scaffold composition and surface properties are key factors in achieving bone tissue regeneration with an adequate osteogenic differentiation of dental mesenchymal cells [47]. In comparison with simple polymeric materials, composite biomaterials reinforced with calcium phosphate ceramics exhibit higher mechanical stability allied to lower degradation. This type of scaffold has also been shown to possess osteoconductive properties to MSCs with the expression of osteoblast-like gene markers [48]. Rodrigues et al. [14] and Salgado et al. [13,49] showed that the integration of nanoHA in scaffolds enabled the recruitment of bone marrow MSCs and boosted their osteogenic differentiation. 

However, fewer studies on dental stem cell application in tissue engineering have introduced hDFMSC to 3D tissue regeneration. Salgado and collaborators [41] cultured hDFMSC in 3D porous scaffolds of collagen–nanohydroxyapatite with phosphoserine (collagen–nanoHA/OPS), an osteogenic inductor. These 3D hDFMSC-loaded collagen–nanoHA/OPS scaffolds showed a higher accumulation of OPN with improved osteogenic differentiation. Carvalho et al. [50] studied biomimetic OPN-enhanced collagen matrices, which showed higher cell proliferation, promoting early MSC osteogenic differentiation and angiogenesis, resulting in a sustained tissue growth response that resulted in a mineralized tissue similar to bone. The results of this study also showed the ability of hDFMSC, cultured within Coll/nanoHA OPN-Fb biocomposite scaffolds, to proliferate and differentiate into osteoblastic cells, with a higher number of cells (higher DNA content) and ALP enzyme activity. DNA extraction quantification showed a higher overall cell proliferation rate into the Coll/nanoHA OPN-Fb biomimetic scaffold (Figure 3). Functional activity tests of hDFMSC also presented favorable results in the OPN-modified scaffold, showing a gradual increase in ALP activity from the early stages of differentiation (7 days) to later osteogenic maturation stages (21 days—Figure 4a). Moreover, in accordance with previous studies performed by Salgado et al. [41], Schwartz et al. [51], and Carvalho et al. [50], all the materials that exhibited decreasing levels of proliferation also showed increasing levels of osteogenic differentiation capacity due to an increase in the ALP activity. This behavior was a positive outcome that should end up leading to normal tissue growth for the proposed clinical application.

The results of the assessment of total protein content showed a global protein increase rate in Coll/nanoHA OPN-Fb scaffolds when compared with the other samples (Figure 4b). This means that hDFMSC are secreting more bone ECM proteins for the matrix mineralization in the 3D scaffold with OPN modification. CLSM images confirmed the highest hDFMSC number of cells within the Coll/nanoHA OPN-Fb scaffolds. Cellular aggregates with a round shape formed a continuous cell layer over the material’s surface with major OPN secretion, thus showing their capacity to differentiate in vitro within the OPN-modified scaffold (Figure 7). Similarly, Mori et al. [44] found that hDFMSC cultured in favorable conditions showed increased ALP activity and produced in vitro mineralized matrix nodules.

Biocompatibility is the main characteristic of biomaterials that will reduce the prolonged inflammatory response and implant encapsulation by fibrotic tissue [52]. To check the inflammatory response of the novel 3D bioconstruct, samples were put in direct and indirect contact with THP-1 monocytes differentiated to macrophages. Our results show that the secretome of undifferentiated hDFMSC (cultured within Coll/nanoHA and basic medium) induced THP-1 macrophages to polarize towards the M1 phenotype (lower percentage, 28.3%) and M2 phenotype (higher percentage, 49.5%). However, the direct interaction of these cells cultured into Coll/nanoHA scaffolds did not show the same behavior; THP-1 macrophages with M1 and M2 polarization were not significantly present when compared with the negative control (less than 17%). Kazimierczak and co-authors (2021) evaluated the THP-1-differentiated macrophages on a chitosan/agarose/nanoHA surface. They observed a higher number of M2-polarized macrophages by morphology analysis that significantly increased the release of anti-inflammatory cytokines (IL4 and TGFβ1) after 7 days [53]. They also observed that these scaffolds had the ability to adsorb large amounts of fibrinogen, and they assumed that M2 macrophage polarization was induced by this specific surface binding. In our study, the macrophages were not cultured on the surface of the biomaterial, and the 2D culture (tissue culture well plate) should have reduced the interaction between them, reducing the macrophage polarization into M1 and M2 phenotypes. 

A major obstacle in the assessment of cell-based therapeutic strategies is the ability to track their interaction with local microenvironments and differentiation into various cell phenotypes as they occur in real time. Currently, the standard methods for determining such metrics are restricted to in vitro studies or limited by histological techniques that rely on the assessment of tissue samples collected from animals sacrificed at specific time points of interest. This study presents an alternative strategy that allows the MSCs’ growth and differentiation within a biomaterial to both be tracked in living animals, repeatedly, in a longitudinal study. Although some of the previous studies used GFP-based transgenic mouse models that exploited an osteoblast lineage-specific promoter to visualize similar events, the techniques that they employed required animal sacrifice for histological analysis, preventing serial follow-up [54]. Alternatively, the presented pre-clinical mouse model expressing the Luc reporter gene under the human dental follicle MSC (GFP-Luc hDFMSC) promoter can be used to track cells that independently and constitutively express GFP for in vitro cell tracking, sorting, and post-imaging histological analysis, and the Luc expression repeatedly and non-invasively in mice. However, such a transduced cellular approach may not be safe enough for imaging in emerging stem cell therapies in humans due to their implantation and possible long-term survival. although, this optical imaging analysis proved to be an optimal tool to evaluate cell therapy efficacy before the patient’s treatment.

## 4. Materials and Methods

### 4.1. Materials

Type I collagen from bovine Achilles tendon (Sigma-Aldrich, St. Louis, MO, USA), nanohydroxyapatite (nanoXIM) aggregates (Fluidinova S.A., Maia, Portugal), 1-ethyl-3-(3-dimethyl aminopropyl) carbodiimide hydrochloride (EDC) and N-hydroxysuccinimide (NHS) (Fluka, Buchs, Switzerland), hydrochloric acid (HCl), Alamar blue dye (resazurin), Fibrinogen, Thrombin, CaCl_2_, NaCl, p-nitrophenol phosphate, Sodium azide, KCl,, Tris(hydroxymethyl)aminomethane (Tris), Human osteopontin (OPN), Phorbol 12-myristate 13-acetate (PMA), Interferon Gamma, Interleukin 10, 40 -6-diamidine-2-phenyl indole (DAPI), formaldehyde 4% and Triton X100, rabbit anti-human osteopontin were purchased from Merck (Darmstadt, Germany); Dulbecco’s modified eagle medium (DMEM), fetal bovine serum (FBS), penicillin-streptomycin and trypsin were purchased from Gibco (Thermo Fisher Scientific, Waltham, MA, USA). The Thermo Scientific™ Pierce™ BCA Protein Assay Kit, Alexa fluorconjugated phalloidin 594 and the Quant-iT™ Picogreen^®^ DNA assay kit were purchased from Invitrogen (Thermo Fisher Scientific, Waltham, MA, USA).

### 4.2. Biomimetic Composite Production

#### 4.2.1. Preparation of Collagen/Nanohydroxyapatite Scaffolds

Coll/nanoHA scaffolds were obtained using a cryogelation technique, as described earlier [13]. Initially, type I insoluble collagen was swelled overnight in 10 mM chloride acid (HCl) solution at 4 °C, and at a concentration of 2% (*w/v*). Then, the dispersion was homogenized (Ultra Turrax T25, IKA, Staufen, Germany) at 11,000 rpm for 1 h. Coll/nanoHA composite cryogel was set with 10 mL of collagen slurry diluted in 4.5 mL of HA (2%) suspended in 10 mM chloride acid (HCl) (final composition, 1:1:0.5 wt). For the preparation of cryogels, 20 mM NHS and 40 mM EDC were added to the collagen slurry and promptly transferred to a syringe (5 mL) that was used as a mold and kept in a freezer at −18 °C for 24 h. Lastly, the scaffolds were lyophilized with a freeze dryer (FreeZone 6, Labconco, Kansas City, MO, USA) at 0.1 bar for 24 h. The samples were cut with a surgical blade with 4 mm width.

#### 4.2.2. Preparation of OPN–Fibrin Hydrogel

A fibrinogen solution (Fb) was obtained by dissolving in Tris-buffered saline (TBS was prepared by mixing 134 mM of NaCl, 2.7 mM of KCl, and 33 mM of Tris, pH of 7.4, at a concentration of 20 mg/mL). 

Human osteopontin (OPN, recombinant, expressed in NSO cells, ≥95%, SDS-PAGE, Merck, France) solution was obtained by dilution in Tris-phosphate-buffered saline (TBS) with a pH of 7.4 at a concentration of 100 µg/mL. The OPN solution (10 or 100 μg/mL solution) was initially mixed with fibrinogen. After cell loading, OPN-Fb was gently mixed in a proportion of 1:2 with the Tb/CaCl_2_ work solution (final concentrations of 6.3 mg/mL of Fb, 10 μg of OPN, 2 National Institute of Health (NIH) U/mL thrombin from human plasma, 2.5 mM CaCl_2_).

### 4.3. Mechanical Characterization 

#### 4.3.1. Rheometer Analysis of OPN–Fibrin Hydrogel

This technique was applied to the characterization of fibrin hydrogels with different concentrations of OPN that could influence the mechanical behavior of the gels. The viscoelasticity of the fibrin hydrogel samples and fibrin (2 NHI U/mL) + OPN (LC and HC) were analyzed using a Kinexus Pro rheometer (Malvern). The fibrin hydrogel samples were punched from the original ones with a cylinder diameter of 4 mm and incubated at 37 °C before testing. All measurements were performed at 37 °C using a plate-and-plate geometry with 4 mm diameter. The samples were compressed to 30% of their original thickness to avoid slippage. Initially, a stress sweep test was performed with a 0.1 Hz frequency to evaluate the linear viscoelastic region (LVR) of the samples. Frequency sweeps between 0.01 and 2 Hz were then performed within the LVR. Three samples of each fibrin hydrogel combination were analyzed, and the average tan delta, storage modulus (G′), and loss modulus (G″) were calculated. The statistical test two-way ANOVA was performed with GraphPad software for the different samples at 0.1 Hz frequency and 1 Hz frequency and displayed in the graphic. 

#### 4.3.2. Dynamic Mechanical Analysis of Coll/nanoHA Scaffold Loaded with OPN–Fibrin Hydrogel

Biomechanical properties of the different proportions of fibrin gel within Coll/nanoHA scaffolds with or without OPN (LC and HC) and the fibrin gel were analyzed with the Tritec2000 dynamic mechanical analyzer (Triton Technology, San Juan Capistrano, CA, USA). This technique applies oscillatory stress to a sample allowing the evaluation of its mechanical response. Thus, it is possible to analyze the viscosity and elasticity of the material from the phase lag and the stiffness (modulus) recovery of the sample. To test the samples, compression cycles under frequencies varying between 0.1 Hz and 15 Hz at 37 °C were used. The sample diameter was previously measured with a Vernier caliper tool, and the samples were incubated at 37 °C before testing. Three samples of each different biocomposite were analyzed. The results (storage modulus, loss modulus, tan delta) were plotted in a graph using an average calculation. 

### 4.4. In Vitro Biological Evaluation

#### 4.4.1. Establishment of Stem Cell Cultures from the Human Dental Follicle (hDFMSC)

To start with, human dental tissue fragments isolated from patient follicle tissue (approved by the Ethical Committee of the University of Porto—50/CEUP/2018, Porto, Portugal) were digested, and hDFMSCs were isolated by adherent culture on plastic tissue culture substrates. After confluence, cells were separated and characterized by flow cytometry and qPCR analysis [41]. Human follicle MSCs were cultured in a basic cell culture medium and kept at 37 °C in a 5% carbon dioxide (CO_2_) atmosphere.

hDFMSCs were cultured in Dulbecco’s Modified Eagle Medium supplemented with 10% fetal bovine serum (FBS) and 1% penicillin/streptomycin (3 × 10^4^ mol/L and 5 × 10^4^ mol/L), maintained at 37 °C and 5% CO_2_. After cell confluence of 90% in the T flask (75 cm^2^; Nunc), cells were washed with PBS solution, detached with trypsin solution (0.5%) at 37 °C for 5 min, and counted using a BRAND^®^ counting chamber BLAUBRAND^®^ Neubauer pattern (Germany). 

#### 4.4.2. Cellular Delivery within OPN–Fibrin Hydrogel

The fibrin hydrogel (with and without OPN) with a total of 3 × 10^5^ cells/scaffold was gently pipetted up and down within each Coll/nanoHA scaffold. As a control, hDFMSCs were centrifuged and concentrated in a small volume (3 × 10^5^ cells/10 μL) and dropped into the scaffold. Subsequently, the scaffolds were placed inside a non-tissue culture 24-well plate for 30 min in a standard incubator (37 °C, 95% humidified air, and 5% *v/v* CO_2_) to allow cell adhesion and hydrogel cross-linking.

Afterward, the wells were filled (1 mL) with basic cell culture medium and incubated at 3, 7, 14, and 21 days (time points). This osteoinductive cell culture medium was prepared with Dulbecco’s Modified Eagle Medium with 10% FBS, 1% P/S, 10 mM of β -glycerophosphate, and 0.1 mM of dexamethasone. The osteoinductive medium was added after 3 days of scaffold incubation. The cell culture medium was changed every 3rd day.

These experiments were aimed at evaluating hDFMSC proliferation and differentiation within the biomaterials, including measuring the total DNA concentration and alkaline phosphate activity, the total protein quantification, RT-qPCR, and human osteopontin immunostaining (osteogenic differentiation).

#### 4.4.3. Cellular Proliferation Assay

DNA content was measured using the Quant-iT™ Picogreen^®^ DNA assay according to the manufacturer’s instructions. After each time point, scaffolds were washed with PBS and then incubated with 0.5 mL of ultra-pure water at 37 °C and 5% CO_2_ for 1 h. Subsequently, scaffolds were placed in a freezer at −20 °C until the end of the experiment and then thawed at room temperature to lyse all the cell membranes. The scaffolds were cut into pieces with scissors and vortexed for 20 s; subsequently, the solution was centrifuged at 2000 rpm for 5 min. The supernatant with the lysed cells was collected (20 μL) and incubated with Picogreen^®^ solution. Lastly, the fluorescence intensity was measured with a microplate spectrofluorometer (SynergyMx, BioTek, Winooski, VT, USA) at 480 and 520 nm excitation and emission, respectively. The results are expressed in nanograms of DNA per mL.

#### 4.4.4. Cellular Differentiation Assay 

The alkaline phosphatase activity (ALP) was measured as a quantitative analysis for early osteogenic characterization. The same supernatant with the lysed cells obtained as described above (Section 4.4.3) was used for the enzyme activity and total protein content protocol. The ALP enzyme activity was followed through substrate hydrolysis, using p-nitrophenol phosphate, in an alkaline buffer solution (pH = 10). After 1 h of incubation at 37 °C, the reaction was suspended by adding NaOH (0.02 M) and the p-nitrophenol was quantified by absorbance at 405 nm using a plate reader (Synergy MX, BioTek, Winooski, VT, USA). Finally, the ALP results were expressed in nanomoles (nmol) of p-nitrophenol produced per minute (min). The ALP activity results were normalized to total protein content and were expressed in nanomoles of p-nitrophenol produced per minute per mg of protein. Total protein content was measured by Lowry’s method (Thermo Scientific™ Pierce™ BCA Protein Assay Kit) with bovine serum albumin used as standard. Results were expressed in mg of protein concentration per ml.

The expression of relevant osteogenic genes was followed at day 21 of culture by real-time quantitative polymerase chain reaction (RT-qPCR). Briefly, total RNA was extracted with a NucleoSpin kit (NucleoSpin RNA, Macherey-Nagel, Dueren, Germany) and reverse-transcribed into complementary DNA (cDNA) with the iScriptTM cDNA Synthesis Kit (BioRad, Hercules, CA, USA), as recommended by the manufacturers. The expression of the target genes was quantitatively determined on RT-PCR equipment (CFX384, BioRad, Hercules, CA, USA) using iTaqTM Universal SYBR^®^ Green Supermix (BioRad, Hercules, CA, USA). All genes were normalized to the reference gene glyceraldehyde 3-phosphate dehydrogenase (GAPDH, Invitrogen, Waltham, Massachusetts, USA) and are described in Table 2. Relative quantification of gene amplification by qPCR was performed using the cycle threshold (Ct) values, and relative expression levels were calculated using the 2^−ΔΔCT^ method. For each qPCR, samples were analyzed in duplicate and three independent experiments were performed.

#### 4.4.5. Confocal Laser Scanning Microscopy (CLSM)

Two samples from each time point were fixed with 4% paraformaldehyde and incubated for 30 min at room temperature. Then, the materials were incubated with 0.1% Triton ×100 solution for 30 min at room temperature and washed twice with 1% bovine serum albumin (BSA) in PBS. The samples were then incubated for 1 h in BSA solution. The cells’ cytoplasm (actin fibers) was stained with Alexa-Fluor-conjugated Phalloidin 594 nm (dilution of 1:200) for 1 h at room temperature and under darkness, and nuclei were stained with DAPI (4′-6-diamidine-2-phenylindole at 0.2%) for 5 min. For human osteopontin immunostaining, an identical protocol was performed for the cell membrane’s permeabilization and to block non-specific binding, as described above. Samples were then incubated with rabbit anti-human osteopontin (AB 1870, Merck, 1:1000) overnight at 4 °C. This procedure was followed by 1 h incubation with Alexa Fluor 488 goat anti-rabbit IgG secondary antibody (1:1000). Samples were subsequently washed, and nuclei were counterstained with 1 μg mL^−1^ DAPI for 10 min at room temperature. All samples were covered by Vectashield. Images were acquired with excitation lasers of 405, 488, and 594 nm and evaluated by Confocal Laser Scanning Microscopy (CLSM, Leica SP2 AOBS SE camera, Leica, Wetzlar, Germany).

#### 4.4.6. Assessing Immune Cell Activation (Macrophages—M1 and M2) in Direct Contact with hDFMSC within Biomimetic Scaffold or Indirect Contact with Exposure to Soluble Factors (Secretome)

THP-1 monocytic cell line was cultured in complete RPMI 1640 medium supplemented with 10% FBS and 1% P/S. In order to induce a macrophage-like phenotype, cells were plated in 24-well plates (0.25 × 10^6^ cells/well) with a medium containing 100 ng/mL of phorbol 12-myristate 13-acetate (PMA) (Sigma) and allowed to differentiate for 48h. To obtain macrophages at a resting state, adherent cells were cultured in a medium without PMA for an additional 24 h period. As a control, and mimicking a pro-inflammatory environment, THP-1 cells were stimulated with Interferon gamma (INF-γ, 20 ng/mL), and consequently a M1 polarization was promoted; macrophages were stimulated with interleukine-10 (IL-10, 10 ng/mL) to induce anti-inflammatory M2 macrophages and incubated for 3 additional days. Differentiated macrophages were incubated with the secretome of hDFMSC culture for 21 days within Coll/nanoHA scaffolds with or without OPN-Fb gel. Cells were harvested with 5mM EDTA-PBS and centrifuged at 300× *g* for 5 min at 4 °C, re-suspended in FACS buffer (PBS 1×, 2% FBS, 0.01% sodium azide), and immunostained with anti-CD86-FITC (Immunotools) and anti-CD163-PE (BD Biosciences Pharmingen, San Diego, CA, USA) were used as M1 and M2 specific markers, respectively, for 20 min at 4 °C in the dark. 

### 4.5. In Vivo Evaluation

#### 4.5.1. Animal Model of Ectopic Intramembranous Ossification (IMO)

Coll/nanoHA scaffolds (cylindrical shape—5 × 4 mm) were seeded with hDFMSC and GFP-Luc hDFMSC cells (3 × 10^5^ cells/scaffold) loaded within OPN/Fibrin hydrogel (as described above—Section 4.2.2) and subcutaneously transplanted into the dorso of each nude, female, 8-week-old mouse (6 animals; i3S animal house, Portugal). All animal experiments were approved by the i3S animal Ethics Committee and were approved by DGAV (Portugal). All tests followed EC guidelines for animal welfare. Researchers involved in animal handling were FELASA accredited and DGAV certified for animal experimentation. Animals were anesthetized with 3–5% isoflurane for induction and 1–2% for surgical procedures that were performed under standard aseptic conditions. A midline incision through the dorsal skin was performed, and three subcutaneous pockets were created, one on the right side (control material—without cells) and two on the left side (scaffolds with cells). The dorsal wound was then closed with surgical staples. IVIS Lumina was used to track by cellular luminescence proliferation and volumetric changes. Briefly, the animals were individually placed in an induction chamber, and anesthesia was induced with 2% isoflurane during in vivo BLI using an IVIS Lumina optical imaging device (Perkin Elmer, Waltham, MA, USA). Animals were given D-luciferin substrate (XenoLight D-Luciferin, 3 mg/100 μL/mouse, Perkin Elmer; Waltham, MA, USA) via intraperitoneal injection for 15 min for the luciferase enzymatic reaction to proceed. BLI was carried out at the acquisition time of 5 min using an open filter set. Bioluminescence signal data were collected from a region of interest (ROI) and were expressed as the average photon count per pixel (cm^2^). Moreover, the scaffolds were immediately imaged after implantation to determine the ROI settings of the luminescence signal. The cell-loaded bioconstructs were implanted and followed once a week for 4 weeks.

After recovery, the mice were caged in pairs and allowed to move in their cages without restriction. They were fed with commercial mice chow and water for 4 weeks ad lib. After the required period of time, mice were euthanized with carbon dioxide asphyxiation.

#### 4.5.2. Histology Analysis

All samples were explanted and fixed in 10% neutralized buffered formalin for three days and then processed for histology. Fixed samples were embedded in paraffin and were sectioned longitudinally with a microtome (5 µm in thickness). The sections were stained with Hematoxylin and Eosin (Vector laboratories, Newark, CA, USA) for light microscopy examination.

#### 4.5.3. Immunohistochemical Analysis

Immunohistochemical analyses were performed to stain the human ECM and cells. The human osteopontin (OPN) was probed after antigen recovery. For this purpose, masked epitopes were exposed by treatment with citrate buffer (pH 9, Sigma-Aldrich, Hamburg, Germany) for 20 min at 97 °C. Sections were incubated with rabbit anti-human osteopontin (AB 1870, 1:500). This procedure was followed by 1 h incubation with Alexa Fluor 594 goat anti-rabbit IgG secondary antibody (1:1000, Invitrogen Molecular Probes, Eugene, OR, USA). All slides were mounted in Vectashield™ with DAPI (Vector laboratories, Newark, CA, USA). Images were obtained using a fluorescence inverted microscope (Axio Imager Z1, Zeiss, Jena, Germany).

### 4.6. Statistical Analysis

Data were presented as mean and standard deviation and analyzed using the two-way ANOVA test (GraphPAD software, Insight Venture Partners, New York City, NY, USA). Differences between groups and time points were considered statistically significant when *p* < 0.05.

## 5. Conclusions

Bone fractures, osteodegenerative diseases, and tumor resection lead to bone defects, and therefore there is an increasing demand worldwide for fracture repair and bone regeneration solutions. The development of new techniques that accelerate the fracture healing process and enhance bone regeneration and remodeling is still a challenge in dealing with appropriate materials integration and bone tissue growth. Those bone regenerative needs have prompted the development of an injectable cell-loaded hydrogel with high bioactivity and a 3D scaffold with enough mechanical strength to allow the biocompatibility, biodegradability, and stability that can induce angiogenesis, as well as to achieve an optimal transport of nutrients, oxygen, and growth factors. To address this challenge, this study developed a biomimetic Coll/nanoHA OPN-Fb scaffold that can deliver patients’ cells and promote the materials’ engraftment into the host tissue, providing a novel solution with considerable advantages over current therapies. This should reduce patient hospital stay and surgery time and prevent post-surgery complications while improving the patients’ quality of life with a more cost-effective healthcare. The 3D bioconstruct will enhance the bone regeneration by creating and maintaining the biological space through the OPN osteoinduction effect. This human protein plays a vital role in the recruitment of cells during the early stage of bone regeneration, increasing cell adhesion and proliferation, as well as promoting early MSC osteogenic differentiation with increased ALP activity (pre-osteoblast-like cell phenotype).

Moreover, this study explored the use of hDFMSC as a clinical alternative source for bone tissue engineering with a controlled and reproducible differentiation behavior. The dental MSCs together with the 3D scaffold Coll/nanoHA with OPN-Fb biomimetic hydrogel promoted an in vitro and in vivo increase in cell proliferation, while the cells remained viable and well distributed within the material structure. In addition, the 3D biocomposite showed physico-chemical characteristics similar to normal bone and an increase in the osteogenic gene expression by the hDFMSC that favor its application in bone tissue engineering. Further studies will focus on these MSCs being tested in pre-clinical animal models with critical bone defects to observe their potential to promote bone tissue regeneration.

## Figures and Tables

**Figure 1 ijms-24-01827-f001:**
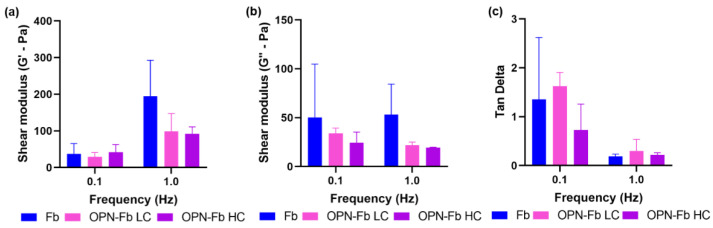
(**a**) Storage modulus (G′), (**b**) loss modulus (G″), and (**c**) tan delta of the fibrin hydrogels with OPN low-concentration (LC) and high-concentration (HC) samples for frequency values of 0.1 Hz and 1 Hz.

**Figure 2 ijms-24-01827-f002:**
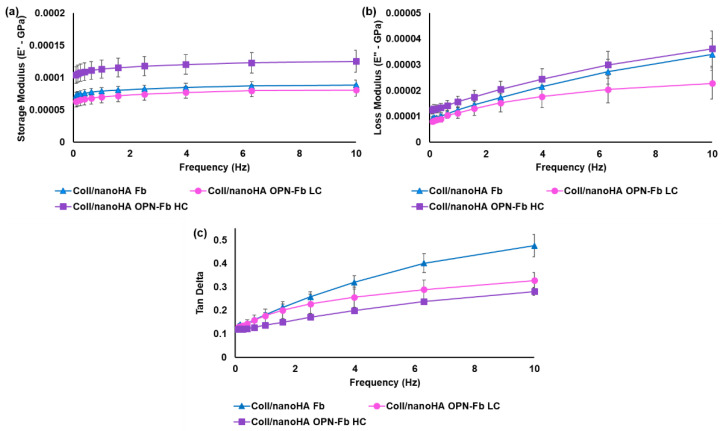
Coll/nanoHA + fibrin gel with low OPN concentration (LC) and high concentration (HC), under dynamic compression solicitation versus frequency (0.1 Hz to 10 Hz). (**a**) Storage modulus (E′), (**b**) loss modulus (E′′), and (**c**) tan delta.

**Figure 3 ijms-24-01827-f003:**
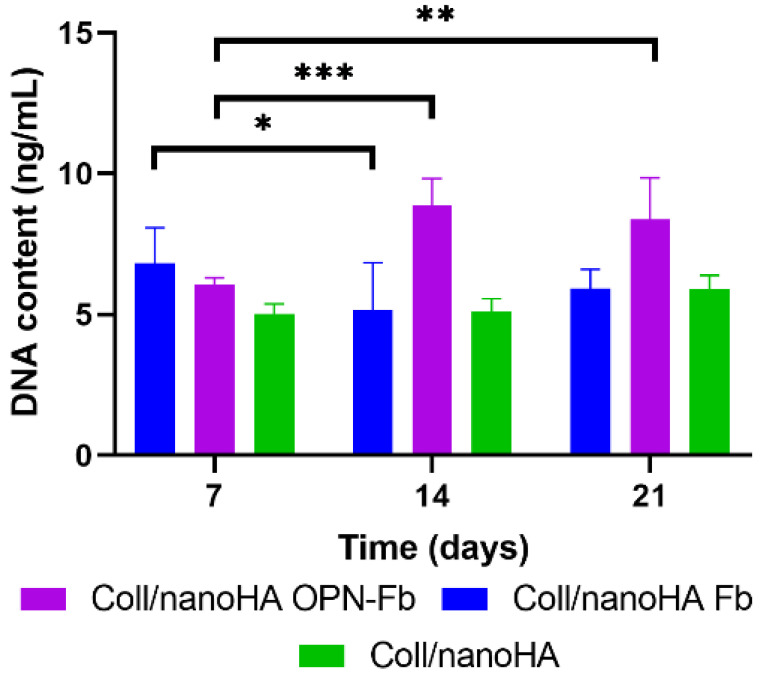
Total DNA content expression of hDFMSC proliferation rate in Coll/nanoHA with Fb, Coll/nanoHA with OPN-Fb, and Coll/nanoHA biocomposite scaffolds for 7, 14, and 21 days of culture in basic medium. Statistical differences between samples from different time points, * *p* < 0.05, ** *p* < 0.01, and *** *p* < 0.001.

**Figure 4 ijms-24-01827-f004:**
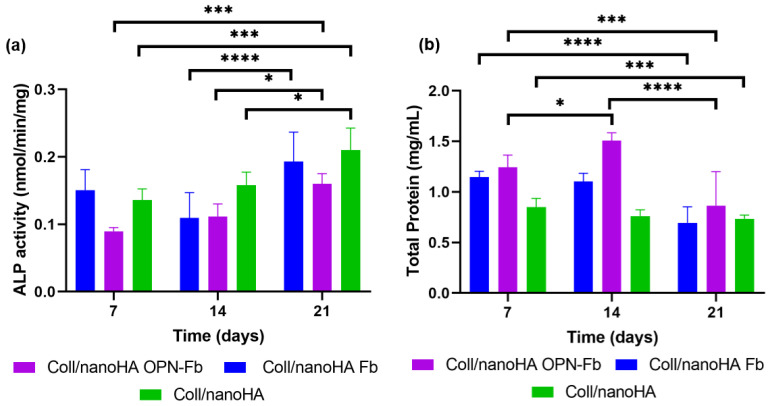
The (**a**) ALP activity and (**b**) total protein secreted by hDFMSC (ECM) in Coll/nanoHA Fb, Coll/nanoHA with OPN-Fb, and Coll/nanoHA biocomposite scaffolds after 7, 14, and 21 days of culture. Statistical differences between samples from different time points, * *p* < 0.05, *** *p* < 0.001, and **** *p* < 0.0001.

**Figure 5 ijms-24-01827-f005:**
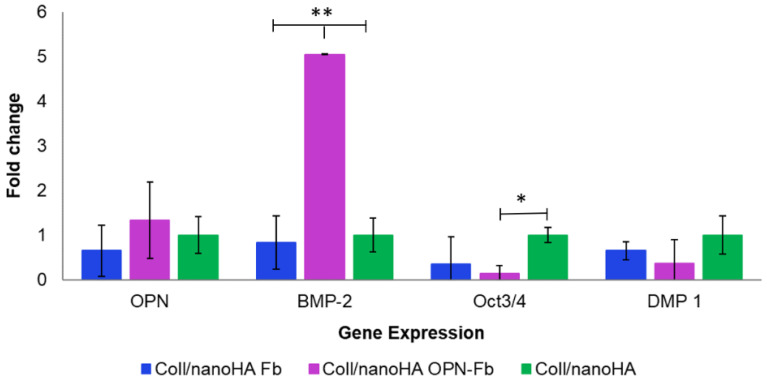
Quantitative real-time polymerase chain reaction (qPCR) for the osteogenic genes (osteopontin and BMP-2) and mesenchymal gene (Oct3/4) and Dentin Matrix Acidic Phosphoprotein 1 (DMP-1) for hDFMSC cultured within the Coll/nanoHA scaffold with or without fibrin hydrogel and osteopontin for 21 days. Quantitative data were calculated by the ΔΔCt method using GAPDH gene expression as an endogenous reference. Samples results were normalized to the undifferentiated cells’ (passage 6) average results. These are represented as fold changes. Statistical analysis, * *p* < 0.05 and ** *p* < 0.01.

**Figure 6 ijms-24-01827-f006:**
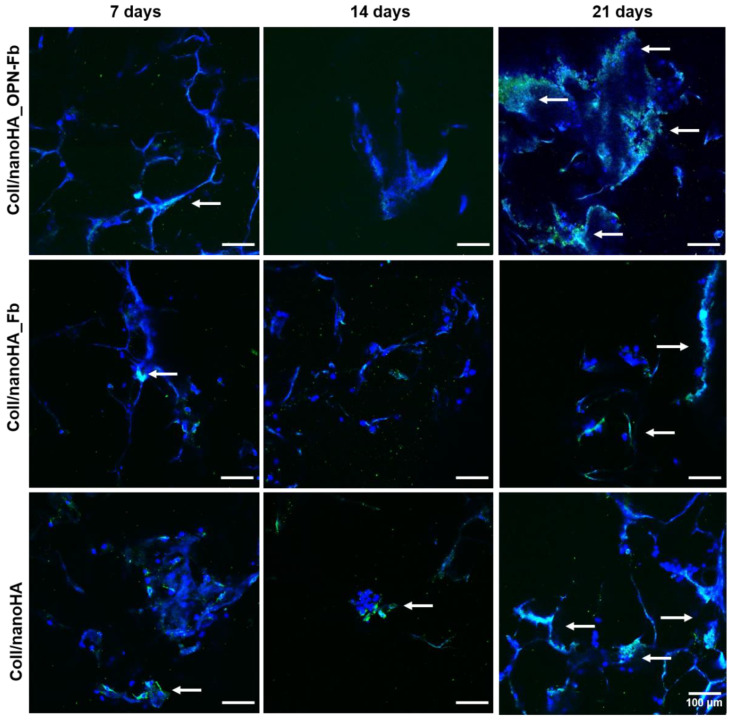
CLSM images showing the human osteogenic ECM (OPN) regarding hDFMSC after 7, 14, and 21 days of culture within Coll/nanoHA scaffold (with or without Fb or OPN-Fb hydrogel). hDFMSC nuclei were stained in blue and human OPN in green (white arrows). Scale bar: 100 μm.

**Figure 7 ijms-24-01827-f007:**
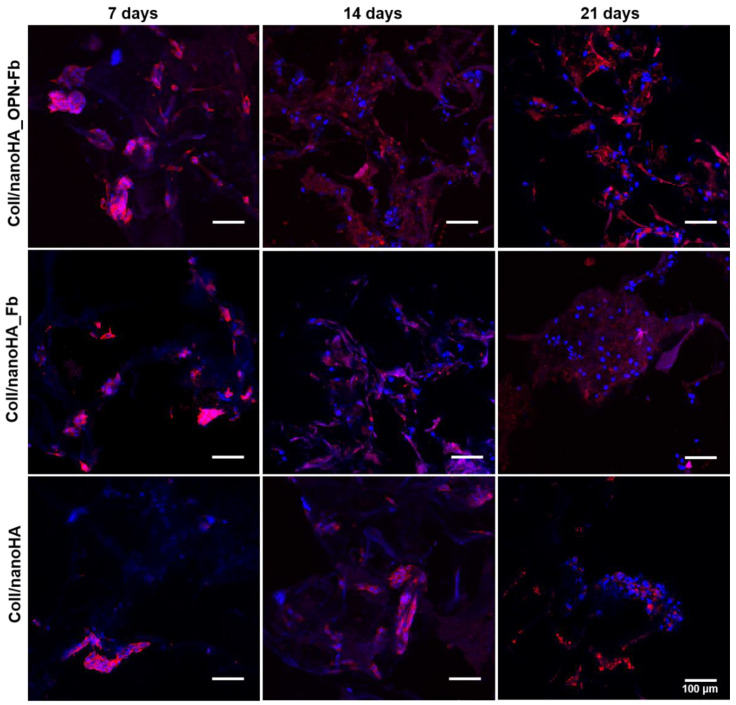
CLSM images showing the morphology of hDFMSCs after 7, 14, and 21 days of culture within a Coll/nanoHA Fb scaffold (with and without OPN) and Coll/nanoHA scaffold. hDFMSCs’ nuclei were stained in blue and cytoplasm (actin filaments) in red. Scale bar: 100 μm.

**Figure 8 ijms-24-01827-f008:**
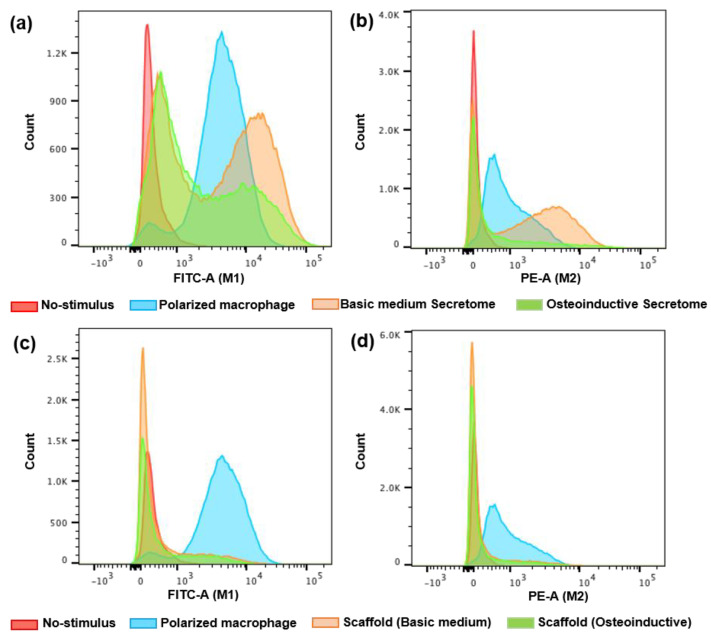
Human monocytes (THP-1) were differentiated into macrophages for 2 days and further polarized for 3 days in the presence of INF-γ (M1) or IL-10 (M2). Surface expression levels of the M1 marker CD86 (FTIC-A) and M2 marker CD163 (PE-A) in hDFMSC secretome (**a**,**b**) or scaffolds in direct contact (**c**,**d**) with 2D cultured macrophages. The macrophage polarization (M1 or M2) was assessed by flow cytometry.

**Figure 9 ijms-24-01827-f009:**
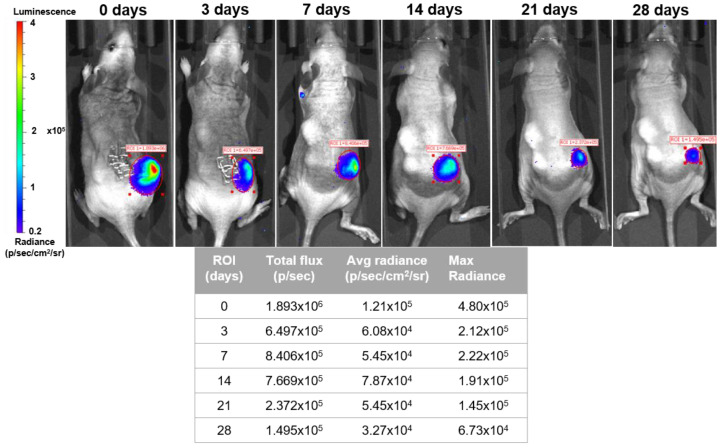
Luminescence optical images of GFP-Luc-hDFMSC-loaded scaffolds after subcutaneous implantation into nude mice for 4 weeks. Table of summary of the image software measurement results.

**Figure 10 ijms-24-01827-f010:**
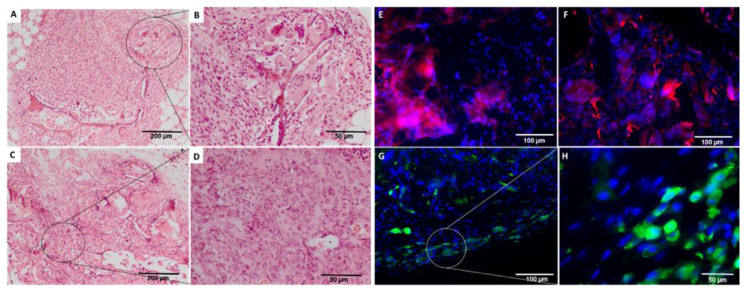
Optical microscopy images of dental MSCs (GFP-Luc hDFMSC—(**C**,**D**,**F**–**H**); hDFMSC—(**A**,**B**,**E**) within Coll/nanoHA scaffolds implanted for 4 weeks. Slides were stained by H&E (**A**–**D**), scale: 200 and 50 µm. Fluorescence microscopy images of human OPN (red) and nucleus (blue) (**E**,**F**), scale: 100 µm; (**G**,**H**) 3D Coll/nanoHA scaffolds with GFP-Luc hDFMSC (GFP+—green and nucleus—blue), scale: 100 and 50 µm.

**Table 1 ijms-24-01827-t001:** Assessment of macrophage activation (M1 and M2), in direct contact with hDFMSC within Biomimetic scaffold or treated with soluble factors (secretome, indirect contact).

Treatment		Macrophage Polarization (%)			
		M0	M1	M0	M2
**Control**	**INF-γ**	5.7	94	x	x
	**No-stimulus**	98	2	x	x
	**IL-10**	x	x	34	66
	**No-stimulus**	x	x	99	0.8
**Indirect contact (secretome)**	**Basic medium**	71.7	28	51	50
	**Osteoinductive medium**	82.8	17	81	19
**Direct contact (scaffold)**	**Basic medium**	83.8	16	86	16
	**Osteoinductive medium**	82.3	18	87	15

**Table 2 ijms-24-01827-t002:** Gene name and respective primers assay ID (Invitrogen) for RT-qPCR.

Gene	Primer Sequence (Forward)	Primer Sequence (Reverse)
**GAPDH**	5′-TAACTGGTAAAGTGGATATTG-3′	5′-GAAGATGGTAGATGGATTTC-3′
**OPN**	5′-ACTCGAACGACTCTGATGATGT-3′	5′-GTCAGGTCTGCGAAACTTCTTA-3′
**BMP-2**	5′-GACGAGGTCCTGAGCGAGTT-3′	5′-GCAATGGCCTTATCTGTGAC-3′
**Oct 3/4**	5′-AGGAGTCCCAGGACATCAAAG-3′	5′-TCGTTTGGCTGAATACCTTC-3′
**DMP-1**	5′-GAGCAGTGAGTCATCAGAAGGC-3′	5′-GAGAAGCCACCAGCTAGCCTAT-3′

## Data Availability

Not applicable.

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
