# Peer review of "Interactions between Dental MSCs and Biomimetic Composite Scaffold during Bone Remodeling Followed by In Vivo Real-Time Bioimaging"

_ijms, 2023, doi:10.3390/ijms24031827_

Round 1

Reviewer 1 Report

The paper by Costa et al. entitled “Interactions between dental MSC’s and biomimetic composite scaffold during bone remodeling followed by in vivo real-time bioimaging” is focused on combining biomimetic materials and dental stem cells to develop a bioengineered construct. The most innovative part was the usage of luminescence/fluorescence for imaging in live animals, better understanding bone behavior. The discussions section is the highlight of the manuscript. Nevertheless, the manuscript can be better written, facilitating the spread of knowledge. Some issues show that the authors could have proof-read the manuscript better, such as the lack of punctuation in Line 19. A few other suggestions are made:

·        Line 34-36. Add reference to the statement: “Commercially available products lack the complex bone structure (architecture and porosity) resulting in biomechanically inferior bone tissue re-35 pair.”

·        Line 105-107. This should be part of the Methods.

·        It is advised to connect better the start of the Results section to the end of the Introduction. The Results section begins with a sub-section about “OPN-Fibrin hydrogel”, which has not been introduced until now. Then, the mechanical properties are quickly described, but no explanation to why they are important is added.

·        The font used in the figures is not consistent. For example, in Figure 8, the font of the axis is different than the font used in the legend.

·        Similarly, the font used in Figure 9 color scale is too small. And, even though the mice figures are well-made, the red inscription around the circles cannot be read. Also, scale bar is lacking.

Author Response

Reviewer #1: The paper by Costa et al. entitled “Interactions between dental MSC’s and biomimetic composite scaffold during bone remodeling followed by in vivo real-time bioimaging” is focused on combining biomimetic materials and dental stem cells to develop a bioengineered construct. The most innovative part was the usage of luminescence/fluorescence for imaging in live animals, better understanding bone behavior. The discussions section is the highlight of the manuscript.

1 - Nevertheless, the manuscript can be better written, facilitating the spread of knowledge. Some issues show that the authors could have proof-read the manuscript better, such as the lack of punctuation in Line 19.

Thanks for your comment, we performed a careful and detailed review of the manuscript text and corrected all the typing errors and mistakes as suggested.

All changes in the manuscript text regarding this information were highlighted with track changes.

A few other suggestions are made:

2 - Line 34-36. Add reference to the statement: “Commercially available products lack the complex bone structure (architecture and porosity) resulting in biomechanically inferior bone tissue repair.”

Thanks for your comment, reference 6 and 7 were added for the statement as suggested.

  1. Alvarez, K.; Nakajima, H. Metallic Scaffolds for Bone Regeneration. Materials 2009, 2, 790-832, doi:10.3390/ma2030790.
  2. Baino, F.; Novajra, G.; Vitale-Brovarone, C. Bioceramics and Scaffolds: A Winning Combination for Tissue Engineering. Frontiers in Bioengineering and Biotechnology 2015, 3, doi:10.3389/fbioe.2015.00202.

Line 36 - “Commercially available products, such as metallic [6] and ceramic [7] materials, usually possess a lack the complex bone structure (architecture and porosity) because of their stiffness and processing requirements resulting in inferior bone tissue integration and repair”.

3 - Line 105-107. This should be part of the Methods.

Thanks for your comment, the text was moved to “Materials and methods” part as suggested.      

All changes in the manuscript text regarding this information were highlighted with track changes.

4 - A) It is advised to connect better the start of the Results section to the end of the Introduction. The Results section begins with a sub-section about “OPN-Fibrin hydrogel”, which has not been introduced until now.

Thanks for your comment, the last paragraph of the introduction was edited to connect better with the results part as suggested.

Line 110-116 – “To address this objective, a 3D bioconstruct based on a collagen/nanohydroxyapatite porous scaffold (Coll/nanoHA) filled with a hydrogel of fibrin modified with osteopontin (OPN-Fb) to carry dental MSCs was developed. This combined approach (scaffold + cell-loaded hydrogel) shall improve the mechanical properties of the graft and provide the necessary pro-regenerative effect to modulate the biological response and could precisely fit the bone defect with fine-tuned adjustment to the surrounding original structure”.

B) Then, the mechanical properties are quickly described, but no explanation to why they are important is added.

Thanks for your comment, the mechanical properties explanation and appropriate references were added to the discussion part as suggested.

Line 329-334 – “Biologically, the increase of the stiffness of the hydrogel has a direct impact on cell behavior. Previous studies showed that high concentrations of fibrinogen and thrombin result in a denser structure affecting normal cell behavior [30]. Thus, lower concentrations of fibrinogen promoted an increase in cell proliferation, while a high concentration of fibrinogen promoted an increase in cell differentiation, but, adversely, could decrease the final mechanical properties (e.g., compressive strength) [31]”.

  1. Noori, A.; Ashrafi, S.J.; Vaez-Ghaemi, R.; Hatamian-Zaremi, A.; Webster, T.J. A review of fibrin and fibrin composites for bone tissue engineering. International journal of nanomedicine 2017, 12, 4937-4961, doi:10.2147/IJN.S124671.
  2. Sproul, E.; Nandi, S.; Brown, A. 6 - Fibrin biomaterials for tissue regeneration and repair. In Peptides and Proteins as Biomaterials for Tissue Regeneration and Repair, Barbosa, M.A., Martins, M.C.L., Eds.; Woodhead Publishing: 2018; pp. 151-173.  5 - The font used in the figures is not consistent. For example, in Figure 8, the font of the axis is different than the font used in the legend.

 Thanks for your comment. All the figures and graphs were reviewed and changed according to the reviewer suggestion (Please check the image in the attachment).

  1. Similarly, the font used in Figure 9 color scale is too small. And, even though the mice figures are well-made, the red inscription around the circles cannot be read. Also, scale bar is lacking.

 Thanks for your comment. A table with all the measurements from the luminescence signal of all time points was added to figure 9. The vertical color scale was improved and is on the left side of the animal’s image. The IVIS Lumina provides an expandable, sensitive imaging system that is easy to use for bioluminescent imaging in vivo. The system includes a highly sensitive CCD camera, light-tight imaging chamber and complete automation. The Living Image software yields high-quality, reproducible, quantitative results incorporating instrument calibration and background subtraction. However, it does not provide a tool for measurement of the area size and a scale bar. The color signal in the field of view (FOV) showed minimum and maximum pixel intensity in the image. The software allows users to track signals during longitudinal studies that vary by many orders of magnitude (Please check the image in the attachment).

Reviewer 2 Report

The authors reported the interaction between dental mesenchymal stem cells and bionic composite scaffolds during bone remodeling and in vivo real-time bioimaging. In total, the paper is well organized and the discussion is convincing. And there are several suggestions/questions are below:

1. The abstract need to describe briefly and precisely for attracting attention of readers. In addition, please check the use of punctuation in line 19 for "imaging in live animals, was designed This in vivo model".

2. Please separate the keywords with ";"

3. The first sentence of the abstract refers to the serious risks posed to patients by the removal of oral and maxillofacial tumours, which are not covered in the text. It is recommended to add relevant content in moderation and cite references.

4. The gold standard for surgical removal of tumours and reconstruction of segmental bone defects is still the use of autogenous bone grafting. Is there anything else that can be done? What are the advantages and disadvantages?

5. The article focuses on the combination of OPN-Fibrin hydrogel with cell therapy as a way to promote bone mucosal healing. Are there other materials that can be combined with cell therapy? How do they compare?

6. Please check for spaces in paragraphs and headings in your article to keep the formatting consistent. For example, 2.2.1, 2.2.2 and part 4 headings are preceded by spaces, while others are not.

7. Please check the format of Fig. 5, there are thin lines in the outer border.

8. There is a large gap under page 6, perhaps resize Figure 6 to fit on one page.

9. The notes on Table 1 are spaced too far apart from the table and the table is not formatted properly, please standardise the diagrams.

Author Response

Reviewer #2: The authors reported the interaction between dental mesenchymal stem cells and bionic composite scaffolds during bone remodeling and in vivo real-time bioimaging. In total, the paper is well organized and the discussion is convincing. And there are several suggestions/questions are below:

  1. The abstract need to describe briefly and precisely for attracting attention of readers. In addition, please check the use of punctuation in line 19 for "imaging in live animals, was designed This in vivo model".

Thanks for your comment, we performed a careful and detailed review of the manuscript text and correct all the typing errors and mistakes as suggested.

The abstract was changed and the following information was added according to the reviewer's suggestion:

“Biologically, the biocomposite based on collagen/nanohydroxyapatite filled with osteopontin-fibrin hydrogel (Coll/nanoHA OPN-Fb) exhibited high cellular proliferation rate with bone extracellular matrix deposition (osteopontin) deposition and increased ALP activity, indicating an early osteogenic differentiation. Thus, the presence of human OPN allowed to enhance hDFMSC adhesion, migration, and spatial distribution within the 3D matrix”.

All changes in the manuscript text regarding this information were highlighted with track changes.

  1. Please separate the keywords with ";"

Thanks for the suggestion, all changes in the manuscript text regarding this information were highlighted with track changes.

  1. The first sentence of the abstract refers to the serious risks posed to patients by the removal of oral and maxillofacial tumours, which are not covered in the text. It is recommended to add relevant content in moderation and cite references.

Thanks for your comment, the information about the oral and maxillofacial tumor was added with all the references in the introduction part as suggested.

Line 35-43 – “Oral cancers are the most derived(???) from the mucosal epithelium in the oral cavity, pharynx and larynx, and are known collectively as head and neck squamous cell carcinoma (HNSCC) [1] that has an incidence around 6 per 100,000 people worldwide (new cases: 377,113/year, deaths: 177,757 in 2020) and is estimated to rise by 47% until 2040 [2]. Screening strategy for HNSCC has not proved to be effective, and pre-malignant lesions usually progress to invasive cancer [1] with bone invasion rate reaching 58% [3]. Clinically, patients are generally treated with surgical resection, followed by adjuvant radiation or chemoradiation [1], from which 66% remain with disfigurements and sustaining scars, requiring restoration of large defects (300,000 new cases/year) [2]”.

  1. The gold standard for surgical removal of tumours and reconstruction of segmental bone defects is still the use of autogenous bone grafting. Is there anything else that can be done? What are the advantages and disadvantages?

As suggested, more information was added:

Line 31 – “The gold standard surgical tumour removal and the treatment for reconstructing segmental bone defects remains the use of bone autografts, particularly to treat large bone defects. Autografts have osteogenic, osteoconductive, and osteoinductive properties, but there are some disadvantages such as the risk of vascular-nervous lesions and increased patient morbidity (second surgery)”.

Line 36 - “Commercially available products, such as metallic and ceramic materials, usually possess a lack the complex bone structure (architecture and porosity) due to their stiffness and processing requirements resulting in inferior bone tissue integration and repair. Another limitation of the commercial biomaterials is the possibility of the eventual release of toxic ions and/or particles, leading to inflammatory response, that may reduce biocompatibility and cause implant loss”.

All changes in the manuscript text regarding this information were highlighted with track changes.

  1. The article focuses on the combination of OPN-Fibrin hydrogel with cell therapy as a way to promote bone mucosal healing. Are there other materials that can be combined with cell therapy? How do they compare?

Thanks for your comment. The OPN-Fibrin hydrogel with dental-MSC therapy innovation relies on the use of a biomimetic hydrogel that had successfully carried mesenchymal cells from dental follicle and increase the materials’ biocompatibility, osteoinduction and tissue ingrowth. Fibrin-based hydrogels possess excellent biocompatibility, contain specific bonding sites for membrane receptors of cell surface, promote cell migration, fibroblast proliferation, and angiogenesis. Also, it has the advantage of being fully injectable and rapid enzymatic degradation that allows cellular delivery and growth. OPN plays a key role in the modulation of osteoblasts, osteoclastic function, and matrix mineralization. Thus, the increase of OPN concentration in the matrix increases the adhesion of osteoblasts and osteoclasts. There are many natural and synthetic materials such as alginate or PEG that are commonly studied to carry cells [1,2]. But, combined with the collagen/nanohydroxyapatite scaffold that was discussed in previous papers [3-6], the OPN-Fibrin hydrogel in this work showed appropriate properties for bone tissue engineering application.

  1. Please check for spaces in paragraphs and headings in your article to keep the formatting consistent. For example, 2.2.1, 2.2.2 and part 4 headings are preceded by spaces, while others are not.

Thanks for your comment, we performed a careful and detailed review of the manuscript text and corrected all the typing errors, configuration, and mistakes as suggested.

  1. Please check the format of Fig. 5, there are thin lines in the outer border.

Thanks for your comment, we performed a careful review of the figure as suggested (Please see the attachment).

  1. There is a large gap under page 6, perhaps resize Figure 6 to fit on one page.

Thanks for your comment, we performed the changes in the manuscript layout as suggested.

  1. The notes on Table 1 are spaced too far apart from the table and the table is not formatted properly, please standardise the diagrams.

Thanks for your comment, we performed a careful review of the tables and standardized the diagrams in all of them as suggested.

References:

  1. Liu, M.; Zeng, X.; Ma, C.; Yi, H.; Ali, Z.; Mou, X.; Li, S.; Deng, Y.; He, N. Injectable hydrogels for cartilage and bone tissue engineering. Bone Res 2017, 5, 17014, doi:10.1038/boneres.2017.14.
  2. Pandolfi, V.; Pereira, U.; Dufresne, M.; Legallais, C. Alginate-Based Cell Microencapsulation for Tissue Engineering and Regenerative Medicine. Curr Pharm Des 2017, 23, 3833-3844, doi:10.2174/1381612823666170609084016.
  3. Rodrigues, S.C.; Salgado, C.L.; Sahu, A.; Garcia, M.P.; Fernandes, M.H.; Monteiro, F.J. Preparation and characterization of collagen-nanohydroxyapatite biocomposite scaffolds by cryogelation method for bone tissue engineering applications. Journal of Biomedical Materials Research Part A 2013, 101A, 1080-1094, doi:https://doi.org/10.1002/jbm.a.34394.
  4. Salgado, C.L.; Barrias, C.C.; Monteiro, F.J.M. Clarifying the Tooth-Derived Stem Cells Behavior in a 3D Biomimetic Scaffold for Bone Tissue Engineering Applications. Frontiers in Bioengineering and Biotechnology 2020, 8, doi:10.3389/fbioe.2020.00724.
  5. Salgado, C.L.; Grenho, L.; Fernandes, M.H.; Colaço, B.J.; Monteiro, F.J. Biodegradation, biocompatibility, and osteoconduction evaluation of collagen-nanohydroxyapatite cryogels for bone tissue regeneration. J Biomed Mater Res A 2016, 104, 57-70, doi:10.1002/jbm.a.35540.
  6. Salgado, C.L.; Teixeira, B.I.B.; Monteiro, F.J.M. Biomimetic Composite Scaffold With Phosphoserine Signaling for Bone Tissue Engineering Application. Frontiers in Bioengineering and Biotechnology 2019, 7, doi:10.3389/fbioe.2019.00206.
